# Plasmalogen Replacement Therapy

**DOI:** 10.3390/membranes11110838

**Published:** 2021-10-29

**Authors:** José Carlos Bozelli, Richard M. Epand

**Affiliations:** Department of Biochemistry and Biomedical Sciences, McMaster University, Hamilton, ON L8S 4L8, Canada

**Keywords:** plasmalogen, plasmalogen-related diseases, degenerative and metabolic disorders, membrane lipid therapy, plasmalogen replacement therapy

## Abstract

Plasmalogens, a subclass of glycerophospholipids containing a vinyl-ether bond, are one of the major components of biological membranes. Changes in plasmalogen content and molecular species have been reported in a variety of pathological conditions ranging from inherited to metabolic and degenerative diseases. Most of these diseases have no treatment, and attempts to develop a therapy have been focusing primarily on protein/nucleic acid molecular targets. However, recent studies have shifted attention to lipids as the basis of a therapeutic strategy. In these pathological conditions, the use of plasmalogen replacement therapy (PRT) has been shown to be a successful way to restore plasmalogen levels as well as to ameliorate the disease phenotype in different clinical settings. Here, the current state of PRT will be reviewed as well as a discussion of future perspectives in PRT. It is proposed that the use of PRT provides a modern and innovative molecular medicine approach aiming at improving health outcomes in different conditions with clinically unmet needs.

## 1. Plasmalogens

The basic structure found in biological membranes is the lipid bilayer. Biological membranes present large lipid compositional diversity because of the presence of qualitatively and quantitatively different molecular lipid species [1]. This lipid chemical heterogeneity is tightly controlled to ensure suitable membrane physical properties and optimal membrane functioning. Plasmalogens, a vinyl-ether subclass of glycerophospholipids, are one of the major lipid components of biological membranes. These lipids are found in a variety of organisms ranging from bacteria to mammals [2]. In mammals, plasmalogen levels are tissue-specific, and their content composes up to 20% of the total membrane lipid [3,4,5]. Because of their high abundance, it is not unexpected that loss of plasmalogens has been associated with several pathologies ranging from inherited to metabolic and degenerative disorders (see Section 3 below).

### 1.1. Chemical Structure

The chemical structure of plasmalogens is similar to their diacyl glycerophospholipids counterparts (Figure 1) [5,6,7]. Plasmalogens differ from their diacyl counterparts by having an alkyl (instead of an acyl) chain attached via a vinyl-ether (instead of an ester) bond to the *sn*-1 position of the glycerol moiety (Figure 1). The presence of a vinyl-ether bond makes plasmalogen different from other ether glycerophospholipids (e.g., plasmanyl phospholipids) (Figure 1). In comparison to the ester bond, the vinyl-ether bond is more hydrophobic and acid/oxidation labile as well as less involved in hydrogen bonds [8].

### 1.2. Membrane Physical Properties

Plasmalogens, due to their different chemical structures, impart different physical properties to membranes in comparison to their diacyl counterparts. For instance, plasmalogens tend to increase lipid packing and membrane thickness, decrease membrane fluidity, and contribute to the formation and stabilization of membrane domains and curved membrane surfaces [7,9,10,11,12,13,14,15,16,17,18,19,20,21,22].

### 1.3. Biological Properties

The fundamental relationship of biology states that the structure (and dynamics) of a molecule or molecular aggregate determines its function. Along this line, it is expected that plasmalogens, by having a different chemical structure, might have a different biological function compared to their diacyl counterparts. Indeed, this is what is found. For instance, one of the main biological functions ascribed to plasmalogens is their ability to function as scavengers of radical species such as reactive oxygen and nitrogen species (ROS/RNS) [23,24,25,26,27,28,29,30,31,32]. In addition, plasmalogens have been suggested to play key roles in signal transduction, including effects on the MAPK/ERK, PI3K/AKT, and PKCδ pathways [33,34,35,36,37]. Plasmalogens can also function by storing signaling molecules as part of the structure of plasmalogens [36,38,39]. More recently, plasmalogen gained increased interest in the study of treatment-resistant cancer due to their involvement in the regulation of lipid peroxidation and ferroptosis (a cell death process triggered by excessive lipid peroxidation) [40,41,42]. However, the molecular mechanisms underpinning the role of plasmalogens in lipid peroxidation and ferroptosis are not completely understood. Finally, plasmalogens have been suggested to play a role in membrane trafficking and viral infection [43,44,45]. In these processes, the biological function of plasmalogens has been proposed to be a consequence of their ability to form and stabilize curved membrane regions, which, in turn, increase membrane remodeling needed during these biological phenomena.

## 2. The Metabolism of Plasmalogens

Plasmalogens’ steady-state levels are a result of the difference between their rate of biosynthesis and rate of degradation. Plasmalogens biosynthesis starts in the peroxisomes and ends in the ER (Figure 2A) [27,46,47,48]. Contrary to the biosynthesis of diacyl glycerophospholipids, whose biosynthesis starts with glycerol-3-phosphate, plasmalogen biosynthesis starts with dihydroxyacetone phosphate (DHAP). In the peroxisomes, DHAP undergoes three sequential reactions to yield 1-alkyl-2-lyso-*sn*-glycero-3-phosphate (AGP), which is transported to the ER, where the final biochemical reactions of plasmalogen biosynthesis take place. Fatty acyl-CoA reductase 1 (Far1) is an enzyme bound to the peroxisomal membrane external (cytoplasmic) surface, which has been proposed to be a rate-limiting reaction in plasmalogen biosynthesis [49,50,51,52].

Plasmalogen degradation could occur either by non-enzymatic or enzymatic biochemical reactions (Figure 2B). The non-enzymatic mechanisms of plasmalogen degradation are chemical in nature and depend on vinyl-ether bond oxidation or hydrolysis; that is, radical or acid attack removes the alkyl chain at the *sn*-1 position of the glycerol moiety [8,27,28]. The enzymatic mechanisms are dependent mainly on the action of phospholipases, each of which could present a different substrate specificity [53,54,55,56,57,58]. In addition, it has been shown that cytochrome c upon oxidative stress can act as a plasmalogenase, releasing the alkyl chain at the *sn*-1 position of the glycerol moiety [59].

**Figure 2 membranes-11-00838-f002:**
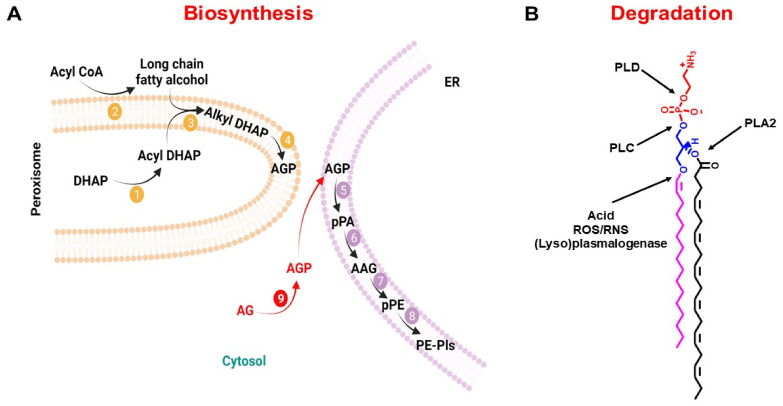
**Metabolism of plasmalogen.** (**A**) Biosynthesis pathway. Initial biochemical reactions start in the peroxisome with the production of 1-acyl-dihydroxyacetone phosphate (1-acyl DHAP) and delivery of 1-alkyl-2-lyso-*sn*-glycero-3-phosphate (AGP) to the ER. In the ER, AGP undergoes four consecutive biochemical reactions to yield PE-Pls. PC-Pls is produced via headgroup transfer and/or remodeling of PE-Pls. Biochemical reactions are catalyzed by 1—GNPAT, 2—Far1, 3—AGPS, 4—AADHAP-R, 5—AAG3P-AT, 6—PAP-1, 7—EPT, and 8—PEDS1. Exogeneous AG crosses the plasma membrane and is phosphorylated in the cytosol (biochemical reaction 9) by an alkyl glycerol kinase before entering the biosynthesis pathway in the ER. (**B**) Degradation pathway. Plasmalogen degradation can occur by the activity of different phospholipases (PLC, PLD, and PLA2) as well as by chemical oxidation/hydrolysis of the vinyl-ether bond. The reader is referred to abbreviations for the names of the lipid intermediates and enzymes found at the end of this article. Schematic representations were generated using Biorender (©BioRender-biorender.com, San Francisco, CA, USA).

## 3. Plasmalogen Changes in Pathophysiological Conditions

Historically, plasmalogens have received little attention compared to various other lipid classes despite their abundance. However, recently this has changed and plasmalogens have started to receive increased attention. This is because of the association between plasmalogens and several pathophysiological conditions. It has been reported that plasmalogen levels are altered in several degenerative and metabolic disorders as well as upon aging. In all these conditions, a common observation is the decrease in the levels of plasmalogens.

In humans, plasmalogens content increases gradually up to 40 years of age, after which it tends to level off and, by the age of 70 plasmalogen content starts to decrease significantly (e.g., there is a 40% decrease in plasmalogen in the serum of individuals more than 70 years old in comparison to younger individuals) [60,61,62]. One of the first associations between plasmalogens and diseases came from studies of peroxisomal deficiency diseases. These are a collection of rare inherited human diseases caused by mutations in genes responsible for peroxisomes biogenesis or function, which include Zellweger syndrome (ZS) and Rhizomelic chondrodysplasia punctata (RCDP) [63,64]. In these diseases, plasmalogen levels are decreased. For instance, in ZS, plasmalogen content has been found to be largely decreased, the extent of which is tissue-specific and can reach up to a 90% decrease in comparison to controls [65]. In RCDP, it has been reported that the decrease in plasmalogen varies with the severity of the phenotype, reaching up to more than 70% reduction in plasmalogen in the most severe phenotype [66]. Decreases in plasmalogen content have also been reported in degenerative and metabolic disorders. In the brain, where the plasmalogen content is the highest, plasmalogen loss has been reported in samples from individuals with different neurodegenerative disorders including Alzheimer’s disease (AD), Parkinson’s disease (PD), and Multiple Sclerosis (MS) [67,68,69,70,71]. Plasmalogen loss has also been reported in cardiometabolic diseases such as Barth Syndrome (BTHS) and coronary artery diseases (CAD) [72,73,74,75,76,77].

In the blood, plasmalogens are found within erythrocyte membranes and lipoproteins [78]. Changes in blood plasmalogens have attracted some interest as a potential biomarker for the diagnosis and prognosis of some pathological diseases [79,80,81,82,83,84,85]. However, in all these cases, comparisons were done with healthy individual controls. In addition, plasmalogen loss has been reported in various diseases (see above), and there is no disease-specific marker reported yet. Hence, changes in blood plasmalogens by themselves should be viewed with caution. A better criterion is to use blood plasmalogen changes with other biomarkers to enhance diagnosis/prognosis accuracy. For instance, it has been shown that changes in PC-Pls containing oleic acid at the *sn*-2 position of the glycerol moiety together with adiponectin and HDL-cholesterol (risk factors for atherosclerosis) in the serum might increase the identification of a proatherogenic state [86].

## 4. Plasmalogen Replacement Therapy (PRT)

A modern and innovative pharmacological approach that started to emerge is membrane lipid replacement [87,88]. A replacement therapy is a pharmacological intervention aimed at restoring the levels of a biological molecule that is deficient in some pathophysiological conditions. Lately, this strategy has attracted increased interest as potentially useful in a variety of pathological conditions including cancer, neurological, and metabolic disorders [89]. Plasmalogen replacement therapy (PRT) is a type of membrane lipid replacement where the strategy relies on the use of small molecules to increase plasmalogen levels with the final goal of improving health outcomes. One of the main advantages of PRT is the possibility to use oral administration. Another one is that the compounds used in PRT usually exhibit no toxicity even at high doses and have been reported to be safe for use in humans [90].

### 4.1. Small Molecules Used in PRT

PRT can be implemented by dietary intervention. For instance, plasmalogens and plasmalogen precursors (intermediates of the plasmalogen biosynthesis pathway) have been found to be enriched in marine animals (e.g., shark liver, krill, mussels, sea squirt/urchin/cucumber, and scallops) as well as in land animals’ meat (e.g., pork, beef, and chicken) (Figure 3) [91]. It has been shown that plasmalogens levels are higher (ranging from 2- to 50-fold, depending on the exact comparison) in livestock and poultry than in fish and mollusk [91]. However, an interesting finding is that the plasmalogens from fish and mollusk have a lower ω-6/ω-3 fatty acid ratio than livestock ones, suggesting the former provide an advantage because of the proposed health benefits of ω-3 fatty acids. While natural sources of plasmalogens bear the potential of providing a dietary plasmalogen supplementation, the decreased bioavailability and the enormous amount of raw material required make it impractical. For instance, scallops have ca. 7.5 μg of plasmalogen/g of muscle. To achieve a common dose of 50 mg/kg, it means that a human with an average weight of 70 kg would need to eat ca. 460 kg of scallops.

To circumvent these problems, purified or synthetic compounds are an attractive alternative to implement PRT as they can be administered at a high dosage. One possibility is to use plasmalogen extracts from natural sources. These plasmalogen extracts tend to be enriched in PE-Pls, and they are often prepared from marine animals (e.g., scallop and sea squirt) or from chicken (Figure 3) [91]. Another possibility is the use of plasmalogen precursors, such as alkylglycerols (AG). AG are lipid intermediates of the plasmalogen biosynthesis pathway (see above) that readily cross the cellular plasma membrane and can enter as a component of the plasmalogen biosynthesis pathway in the ER after being phosphorylated in the cytosol (Figure 2A) [2,94]. AG of different chain lengths have been used, the most common ones being 1-O-hexadecyl-*sn*-glycerol (HG, 16:0-AG), 1-O-octadecyl-*sn*-glycerol (OG, 18:0-AG), and 1-O-octadecenyl-*sn*-glycerol (OeG, 18:1-AG) (Figure 4) [3]. In mammals, oral administration of purified plasmalogens shows an extensive breakdown in the intestinal mucosal cells, while administration of AG leads to complete absorption by the intestine without cleavage of the ether bond, likely because the vinyl-ether bond in plasmalogens is more acid/oxidation labile than the ether bond in AG [95]. In the intestinal mucosal cells, the majority of AG is metabolized into plasmalogens (specifically PE-Pls), while a small fraction is transported to the liver, where it is catabolized [90]. Plasmalogens are transported from the intestine and liver to other organs, but it seems that they do not cross the blood-brain barrier and are not transported across the placenta (from mother to fetus) [90,96]. The use of synthetic analogs of plasmalogens in PRT has also been described, e.g., PPI-1011 (an alkyl-diacyl plasmalogen precursor with DHA at the *sn*-2 position), PPI-1025 (an alkyl-diacyl plasmalogen precursor with oleoyl at the *sn*-2 position), and PPI-1040 (a PE-Pls analog with a proprietary cyclic PE headgroup) (Figure 4) [97,98,99]. PPI-1011 is bioavailable in rabbits and leads to an increase in PE-Pls levels in circulation. In addition, its metabolites can cross both the blood-retina and the blood-brain barriers [97]. PPI-1040 leads to a greater increase in plasmalogen level and bioavailability than PPI-1011 [99]. Indeed, contrary to PPI-1011, PPI-1040 has been reported to be intact during digestion, absorption, and circulation, possibly explaining its greater efficacy [97,99].

### 4.2. In Vitro PRT Studies

PRT has been studied in different clinical settings from cells to animals and humans (Table 1). In cells, PRT has been shown to be successful in increasing plasmalogen levels and alleviating some disease-related phenotypes (Table 1). For instance, in lymphoblasts derived from BTHS patients, adding HG to the media 20 h before collecting the cells led to a significant increase in PE-Pls levels [100]. In this study, an increase in cardiolipin levels and an improvement in mitochondrial fitness were also found, indicating the potential benefit of PRT to improve health outcomes of BTHS patients. For ZS, it was shown that administration of HG to fibroblasts derived from ZS subjects results in increased PE-Pls levels and decreased β-adrenergic receptor stimulation by isoproterenol (an agonist), a result of a reduced number of receptors induced by HG treatment [101]. In lymphocytes derived from RCDP patients, administration of PPI-1011 increased PE-Pls levels [97]. In addition, administration of purified PE-Pls to neurons promoted differentiation, with the greatest effect coming from PE-Pls purified from a marine mollusk (*M. edulis*) rather than bovine brain, possibly due to differences in lipid molecular species [102]. Scallop-purified PE-Pls was shown to have anti-inflammatory properties as indicated by reduction of microglia activation, Toll-like receptor 4 (TLR4) endocytosis, and caspase activation [33]. In neurons, chicken purified PE-Pls has anti-apoptotic properties, as indicated by a decrease in caspase activation and activation of PI3K/AKT and MAPK/ERK signaling pathways [103]. Likewise, eicosapentaenoic acid (EPA)-enriched PE-Pls also showed anti-apoptotic properties in primary cultured hippocampal neurons by upregulating anti-apoptotic proteins and downregulating apoptotic proteins [104]. In addition, in an isolated rat heart, administration of HG decreased myocardial ischemia/reperfusion injury [30].

### 4.3. In Vivo PRT Studies

In animals, PRT was also successful in increasing plasmalogen levels and ameliorating some disease-related phenotypes (Table 1). In a mouse model of RCDP (GNPAT knockout), OG administration for 2 months replenishes cardiac levels of PE-Pls and normalized cardiac impulse [105]. In another mouse model of RCDP (Pex7 knockout), OG increased PE-Pls levels in peripheral tissues and nervous tissue as well as improving nerve conduction [106]. In this study, PRT stopped the progression of pathology in testis, adipose tissue, and eyes. In the same Pex7 knockout mice, administration of PI-1040 increased levels of PE-Pls in the plasma, erythrocytes, liver, small intestine, skeletal muscle, and heart, but not in brain, lung, or kidney [99]. In this study, PRT reduced the hyperactive behavior of Pex7 knockout mice. Interestingly, PPI-1011 did not show the same effects as PI-1040 treatment, suggesting that the latter is a better candidate for use in future investigations of RCDP. Indeed, in 2019 the FDA (Food and Drug Administration) granted PPI-1040 orphan drug designation for treatment of RCDP.

**Table 1 membranes-11-00838-t001:** Summary of the use of PRT in different clinical settings.

Pathological Condition	Model	PRT Compound	Administration	Dosage	Time	Effects	
						Lipids	Phenotype
	**In Vitro**						
BTHS [100]	Lymphoblasts from patients	HG	Added to culture medium	50 μM	20 h	Increased PE-Pls and CL	Restored mitochondrial membrane potential
ZS [101]	Fibroblasts from patients	HG	Added to culture medium	63 μM	24 h	Increased PE-Pls	Decreased β-adrenergic signaling
AD [33]	Neuroinflammation in BV2/primary microglial cells	Scallop-purified PE-Pls	Added to culture medium	6 μM	12 h	Not reported	Inhibition of LPS-mediated TLR4 endocytosis and downstream caspase activation
AD [103]	Neuronal apoptosis in Neuro-2A/primary hippocampal neurons	Chicken skin-purified PE-Pls	Added to culture medium	6–24 μM	72 h	Not reported	Inhibition of caspase-3/9 and activation of PI3K/AKT and MAPK/ERK signaling pathways
AD [104]	Neuronal apoptosis in primary hippocampal neurons	PE-Pls (EPA-enriched)	Added to culture medium	6–72 μM	24 h	Not reported	Upregulation of anti-apoptotic proteins and downregulation of pro-apoptotic proteins
Myocardial Ischemia/Reperfusion Injury [30]	Isolated rat heart	HG	Perfusion	50 μM	15 min	Not reported	Reduced Myocardial ischemia/reperfusion injury
	**In Vivo**						
PD [98]	MPTP-treated mice	PPI-1025	Oral administration	10–200 mg/kg	10 days	Increased PE-Pls with octadecyl alkyl chain in serum	Prevention of MPTP-induced decrease in dopamine/serotonin
RDCP [99]	Pex7^hypo/null^ mice	PPI-1040	Oral administration	50 mg/kg	4 weeks	Increased PE-Pls in plasma, erythrocyteand peripheral tissue, but not in brain, lung, or kidney	Normalized hyperactive behavior
AD [33]	Triple transgenic mice expressing mutant APP, PS1, and Tau	Scallop-purified PE-Pls	Oral administration	133 nM	15 months	Not reported	Reduced endocytosis of TLR4 of the brain cortex
AD [104]	Aβ_42_-treated rats (injected in the brain)	PE-Pls (EPA-enriched)	Administered by gavage	150 mg·kg^−1^· day^−1^	26 days	Not reported	Suppressed neuronal loss and enhanced BDNF/TrkB/CREB signaling
RCDP [105]	GNPAT knockout mice	OG	Oral administration	2% *w*/*v*	2 months	Increased cardiac PE-Pls	Normalized cardiac conduction velocity
RCDP [106]	Pex7 knockout mice	OG	Oral administration	2% *w*/*v*	2-4 months	Increased PE-Pls in peripheral and nervous tissues	Stopped progression of pathology in testis, adipose tissue, and eyes; nerve conduction in peripheral nerves improved
AD [107]	Mice (systemic LPS-induced neuroinflammation)	Chicken-breast-purified PE-Pls	Intraperitoneal injection	20 mg/kg	7 days	Suppressed PE-Pls reduction in the PFC and hippocampus	Attenuated microglia activation and accumulation of Aβ proteins
AD [108]	Aβ_42_-treated rats (injected in the brain)	PE-Pls (EPA-enriched)	Administered by gavage	150 mg·kg^−1^· day^−1^	26 days	Not reported	Alleviated Aβ-induced neurotoxicity by inhibiting oxidative stress, neuronal injury, apoptosis, and neuro-inflammation
AD [109]	Aβ-infused rats	Ascidian-purified PE-Pls	Oral administration	209 μmol·kg^−1^·day^−1^	4 weeks	Increased PE-Pls in plasma, erythrocyte, and liver	Improvement in reference and working memory-related learning abilities
PD [110]	MPTP-treated mice	PPI-1011	Oral administration	5–50 mg·kg^−1^	10 days	Increased PE-Pls with octadecyl alkyl chain in serum	Prevention of MPTP-induced decrease in dopamine/serotonin
PD [111]	MPTP monkeys	PPI-1011	Oral administration	50 mg·kg^−1^	12 days	Increased serum PE-Pls	Decreased L-DOPA-induced dyskinesias
PD [112]	MPTP-treated mice	PPI-1011	Administered by gavage	10–200 mg·kg^−1^	10 days	Increased plasma PE-Pls	Prevented loss of tyrosine hydroxylase (TH) expression and reduced the infiltration of macrophages in the gut
PD [112]	MPTP monkeys	PPI-1011	Oral administration	25 mg·kg^−1^	28 days	Not reported	Reduced L-DOPA-induced dyskinesia
Atherosclerosis [113]	Hamster (High-fat diet)	Sea urchin-purified PE-Pls	Dietary supplementation	0.03%	8 weeks	Decreased total cholesterol and LDL-cholesterol	Reduced atherosclerotic lesion area, attenuated the degree of liver steatosis
Atherosclerosis [104]	LDL receptor-deficient mice (High-fat diet)	Sea cucumber-purified PE-Pls	Dietary supplementation	0.01%	8 weeks	Decreased total cholesterol and LDL-cholesterol; Increased total neutral sterol and bile acids in feces	Reduced atherosclerotic lesion area
Cardiac remodeling [114]	Dominant negative PI3K (small heart) and overexpression of mammalian sterile 20-like kinase 1 (dilated cardiomyopathy) transgenic mice	OG	Dietary supplementation	2 g·kg^−1^· day^−1^	16 weeks	Increased PE-Pls in the heart	No effect on heart function and size
Cancer [115]	Grafted tumors in mice	AG purified (from shark liver oil)	Dietary supplementation	25 mg·day^−1^	10 days	Decreased plasmalogen content in tumor	Decreased growth, vascularization, and dissemination of Lewis lung carcinoma
	**Clinical Trials**						
Peroxisomal disorder [90]	3 human subjects with low DHAT-AT activity and erythrocyte PE-Pls	OG	Ether lipid suspension	5–10 mg/kg^−1^· day^−1^	27–43 month	Increased erythrocyte PE-Pls	Improvement in nutritional status, liver function, retinal pigmentation, and motor tone
Peroxisomal disorder [116]	2 human subjects with low DHAT-AT activity and erythrocyte PE-Pls	OG	Ether lipid suspension	20 mg/kg^−1^· day^−1^	3–18 months	Increased erythrocyte PE-Pls	Improved growth, muscle tone, general state of awareness
Mild-AD and mild cognitive impairment [117]	Multicenter, randomized, double-blind, placebo-controlled clinical trial with 328 subjects with 20–27 points in MMSE-J and ≤5 points in GDS-S-J	Scallop-purified PE-Pls	Oral administration	1 mg/day	24 weeks	Treatment had lowered the decrease in plasma PE-Pls	No significant differences in primary and secondary outcomes. Subgroup analysis of mild-AD patients, showed improvement in WMS-R (secondary outcome) in females and those aged below 77 years
Mild forgetfulness [118]	Randomized, double-blind, placebo-controlled clinical trial with 50 adult volunteers	Ascidian-purified PE-Pls	Dietary supplementation	1 mg/day	12 weeks	Not reported	Increased score in composite memory (sum of verbal and visual memory scores)
Metabolic disease [118]	Randomized, double-blind, placebo-controlled cross-over clinical trial with 10 (obese or overweight) subjects	Shark liver oil-purified AG	Oral administration	4 g/day	3 weeks treatment/ 3 weeks washout/ 3 weeks placebo (and vice versa)	Increased in PE-Pls and ether lipids in plasma and white blood cells	Decreased plasma levels of total free-cholesterol, triglycerides, and C-reactive protein
Hyperlipidemia/Metabolic disease [119]	17 subjects with obesity and hyperlipidemia	Myo-inositol	Oral administration	5 g/day in week 1 and 10 g/day in week 2	2 weeks	Increased plasma PC-Pls	Decreased in atherogenic cholesterol, including small dense LDL
PD [71]	10 subjects with PD	Scallop-purified PE-Pls	Oral administration	1 mg/day	24 weeks	Increased PE-Pls in plasma and erythrocyte membranes	Improvement clinical symptoms (as evaluated by PDQ-39)

There has also been some interest in the use of PRT in animal models of neurological and metabolic disorders (Table 1). For instance, in mice models of AD, oral administration or intraperitoneal injection of purified PE-Pls led to neuroprotection by attenuating neurotoxicity and neuroinflammation in the brain as indicated by a reduced microglia activation, reduced accumulation of Aβ peptides, decreased neuronal apoptosis, and activation of brain-derived neurotrophic factor/tropomyosin receptor B/cAMP response element-binding protein (BDNF/TrkB/CREB) signaling pathway, and inhibition of oxidative stress [33,104,107,108]. In addition, in cognitive-deficient rats (a model for AD), oral administration of purified PE-Pls led to an increase in PE-Pls levels in the plasma, erythrocyte, and liver, as well as improved reference/working memory-related learning abilities [109]. While an increase in total levels was not observed in the brain, the cerebral cortex became enriched with the major molecular species of the purified PE-Pls, that is, 18:0/22:6-PE-Pls. In rats, the co-administration of myo-inositol and ethanolamine has been shown to increase cerebellum PE-Pls levels as well as decrease cortex ATP and cerebellar oxidative stress [120,121]. In animal (mouse and monkey) models of PD (treated with 1-methyl-4-phenyl-1,2,3,6-tetrahydropyridine, MPTP), oral administration of PPI-1011 increased PE-Pls levels in serum, displayed neuroprotective and anti-inflammatory properties, reversed dopamine and serotonin loss, and showed antidyskinetic activity [71,98,110,111,112,122]. A similar result was obtained with PPI-1025 treatment, suggesting that the effect is independent of the acyl chain at the *sn*-2 position of the glycerol moiety and likely dependent on the vinyl-ether bond or on the entire plasmalogen molecule [98]. In hamster and mouse models of atherosclerosis, oral administration of PE-Pls decreased atherosclerosis lesion and total cholesterol and LDL cholesterol in serum [113,123]. In transgenic mice with small (due to depressed PI3K signaling) or failing hearts (due to dilated cardiomyopathy), OG treatment increased both PC-Pls and PE-Pls levels but did not impact heart size or function [114]. PRT had also shown antitumor properties. In mice, oral administration of shark liver oil (enriched in AG) or shark-liver-oil-purified AG decreased tumor growth, vascularization, and dissemination [115].

### 4.4. Clinical Trials

In humans, PRT has been investigated in subjects with peroxisomal, neurodegenerative, and metabolic disorders (Table 1). Two independent early studies of PRT with humans showed that oral administration of OG to individuals with peroxisomal disorders (characterized by decreased GNPAT activity and decreased levels of PE-Pls in erythrocyte membranes) led to the restoration of erythrocyte PE-Pls content and clinical (growth, muscle/motor tone, general state of awareness, liver function, and retinal pigmentation) improvement [90,116]. In an intention-to-treat analysis, oral administration of scallop-purified PE-Pls to individuals with mild AD and mild cognitive impairment did not show a significant difference from the control (placebo) group in primary (Mini mental state examination-Japanese, MMSE-J) or secondary (Wechsler Memory Scale-revised, WMS-R, Geriatric depression scale-short version-Japanese, GDSSV-J, and plasma erythrocyte PE-Pls levels) outcomes [117]. However, in another study, oral administration of scallop-purified PE-Pls to individuals with mild cognitive impairment, mild-to-severe AD, and PD led to significant increase in plasma and erythrocyte PE-Pls levels as well as cognitive function [84]. Likewise, in subjects with mild forgetfulness, sea-squirt-purified PE-Pls showed improved cognitive function as evaluated by visual and verbal memory scores [116]. Individuals with PD treated with scallop-purified PE-Pls showed increased PE-Pls content in the plasma and erythrocyte membranes [71]. Improvement was also noted in PD clinical symptoms as evaluated by Parkinson’s disease questionnaire-39 (PDQ-39, a self-report questionnaire used to monitor health status of physical, mental, and social domains). In individuals with features of metabolic disease, oral administration of shark liver oil-purified AG led to increased levels of PE-Pls in plasma and circulatory white blood cells, as well as decreased total cholesterol, triglycerides, and C-reactive protein in the plasma. In subjects with hyperlipidemia, myo-inositol treatment led to an increase in PC-Pls and a decrease in atherogenic cholesterol in the serum, suggesting the potential benefit for individuals with metabolic syndrome [119].

## 5. Future Perspective for PRT

PRT provides a new, modern, and innovative pharmacological and nutritional approach aiming at improving health outcomes in clinically unmet needs of local, national, and global importance. Early reports of the use of PRT in humans started over 3 decades ago; however, an increased interest has only started in recent years, mainly due to the increased number of reports describing the association between plasmalogen loss and several pathological diseases. While encouraging, the results obtained in different clinical settings with PRT compounds (see above), PRT still needs further investigation to be brought to the clinic. One of the challenges the area faces is to find strategies to markedly increase plasmalogen levels in the brains. In that regard, the co-administration of myo-inositol and ethanolamine seems a promising avenue [120,121]. It has also been shown that drug-encapsulated nanoparticles are better at transposing the blood-brain barrier if the nanoparticles are modified on the surface with groups that are recognized by blood-brain barrier receptors [124]. The encapsulation of PRT compounds within these systems might provide a strategy to increase plasmalogen levels in the brain and, consequently, improve health outcomes for subjects with neurological disorders. Another challenge is to find strategies that target specific molecular species of plasmalogens. Most of the compounds used seem to increase mainly PE-Pls levels. It would be interesting to find small molecules that specifically increase PC-Pls levels, which might be important for heart-related diseases, as PC-Pls is enriched in the heart. In addition to the headgroup, there are aliphatic chains. Regarding those, most of the studies have focused on the nature of the acyl chain at the *sn*-2 position of the glycerol moiety, while a limited amount of work targeted the nature of the alkyl chain. Future work would benefit by understanding how to increase specific molecular species of plasmalogens. Finally, there is a lack of investigation of the combined action of both PRT and antioxidants. Plasmalogens are lipophilic antioxidants that are anchored to the membrane, which makes them particularly effective in preventing lipid oxidation and consequent membrane damage. However, upon oxidative stress, ROS/RNS in the aqueous environment might be inaccessible to plasmalogens. Hence, the co-administration of PRT compounds and water-soluble antioxidants should present synergistic effects, which would lead to enhanced health outcomes. Indeed, it has been shown that the co-administration of lipids and antioxidants is beneficial for certain clinical disorders [88]. While PRT is in its infancy, the findings so far are encouraging. In the future, more systematic investigations will allow the design of better, more potent strategies for PRT.

## Figures and Tables

**Figure 1 membranes-11-00838-f001:**
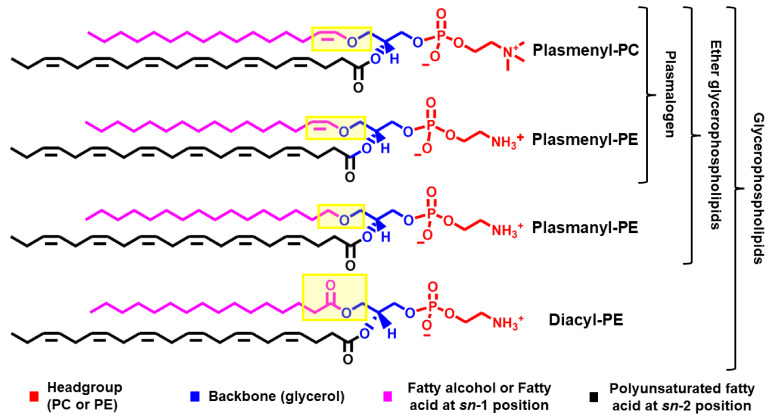
**A comparison between the chemical structure of some glycerophospholipids.** From top to bottom, the lipids are plasmenyl-PC (PC-Pls), plasmenyl-PE (PE-Pls), plasmanyl-PE, and diacyl-PE. The top two lipids differ in the nature of the headgroup (PC vs. PE). PC-Pls and PE-Pls are the major plasmalogen species in mammals. The bottom three lipids differ in the nature of the chemical bond linking the aliphatic chain (pink) to the *sn*-1 position of the glycerol (blue) moiety (highlighted in yellow), where PE-Pls has a vinyl-ether bond, plasmanyl-PE has an ether bond, and diacyl-PE has an ester bond.

**Figure 3 membranes-11-00838-f003:**
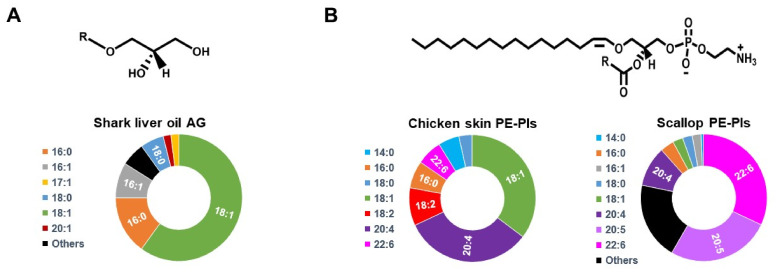
**Plasmalogen and plasmalogen precursors in natural sources.** (**A**) Alkylglycerols (AG) are enriched in shark liver oil. On top is a generic chemical structure of an AG. In the bottom the alkyl chain distribution of shark liver oil-purified AG (from [92]). (**B**) PE-Pls is the major plasmalogen found in chicken skin and scallops. On top is a generic chemical structure of PE-Pls. On the bottom is the acyl chain distribution of chicken-skin- and scallop-purified PE-Pls (from [71,93]).

**Figure 4 membranes-11-00838-f004:**
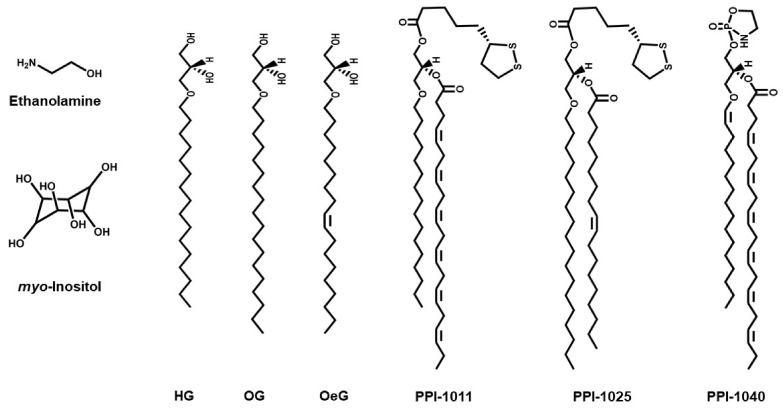
**Chemical structure of compounds used in PRT.** HG, 1-O-hexadecyl-*sn*-glycerol (16:0-AG); OG, 1-O-octadecyl-*sn*-glycerol (18:0-AG); OeG, 1-O-octadecenyl-*sn*-glycerol (18:1-AG); PPI-1011, an alkyl-diacyl plasmalogen precursor with DHA at the *sn*-2 position; PPI-1025, an alkyl-diacyl plasmalogen precursor with oleoyl at the *sn*-2 position; and PPI-1040, a PE-Pls analog with a proprietary cyclic PE headgroup.

## Data Availability

Data used in this article were obtained from the scientific literature; references given below.

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
