# Peer review of "Plasmalogen Replacement Therapy"

_membranes, 2021, doi:10.3390/membranes11110838_

Round 1

Reviewer 1 Report

This manuscript gives an overview of the chemical structure, physical and biological properties, and metabolism of plasmalogen. It then reviews the studies on the association between plasmalogens and diseases, as well as the recent progress of plasmalogen replacement therapy. The manuscript is easy to read. The relevant literature is reviewed thoroughly. Only one minor issue should be resolved: the strange symbol in the unit of dosage (potentially introduced by automatic formatting) needs to be corrected throughout the manuscript and table 1.

Author Response

Reply to reviewers

Here we will address all the points of the reviewers concerning our manuscript entitled “Plasmalogen Replacement Therapy” submitted for publication to the journal “Membranes”.

We would like to thank the reviewers for their comments. We think that by addressing these points our manuscript will reach a broader audience.

Reviewer #1: This manuscript gives an overview of the chemical structure, physical and biological properties, and metabolism of plasmalogen. It then reviews the studies on the association between plasmalogens and diseases, as well as the recent progress of plasmalogen replacement therapy. The manuscript is easy to read. The relevant literature is reviewed thoroughly. Only one minor issue should be resolved: the strange symbol in the unit of dosage (potentially introduced by automatic formatting) needs to be corrected throughout the manuscript and table 1.

We thank the reviewer. We have corrected the symbols, which was introduced by automatic formatting as suggested. Please see revised version.

Reviewer 2 Report

Line 17 and Line 43: According to literature, plasmalogens accounted for about 20% of the total phospholipids. So, I am afraid the word "major" may be too strong. "one of the major" could be better.

Line 48: Please provide references.

Line 61: The authors emphasized the characteristics of "vinyl-ether" and showed the figure of PlsEtn, PakEtn, and PE. But as it is the first figure of the review, I recommend the authors consider all the types of plasmalogens, for example, PlsCho. 

Line 51~249: "A – Chemical structure", "B – Membrane physical properties", "C – Biological properties", "The metabolism of plasmalogens", "Plasmalogen changes in pathophysiological conditions". Frankly speaking, they have been summarized in previous reviews. So, it is not interesting to talk at length about them. Please compact the texts, and let your main point "PRT".

Line 220: Please make the "background" concise. The authors do not need to explain "what is AD/PD", especially, there is an unknown vortex symbol after the word "amyloid" and before the word "synuclein" (and otherwhere in the whole manuscript).

Line 252: What is the definition of "replacement", and what are the differences between "replacement therapy " and simply "therapy/treatment"? Oral administration could not prove it connected with "replacement". I strongly suggest the authors be very careful about using this word unless there is strong evidence.

Line 265: What are "plasmalogen precursors"? I guess the authors did not explain it until Line 288.

Line 252: If possible, please separate this part into several sub-sections, making the main body of this review a "stronger sense of layers".

Page 17 (by the way, there is no line number), "V. Future perspective for PRT": I still feel it is not enough to claim "plasmalogen replacement". 
Suppose, "plasmalogens are oxidants and easy to be oxidized, the anti-oxidative effect of exogenous plasmalogens improves the body conditions, and the endogenous plasmalogens are protected against oxidative stress" -> this is a possible theory, but cannot be described as "replacement"!

Author Response

Reviewer #2: Comments and Suggestions for Authors

Line 17 and Line 43: According to literature, plasmalogens accounted for about 20% of the total phospholipids. So, I am afraid the word "major" may be too strong. "one of the major" could be better.

We agree and we have reworded the text (please see below).

Former lines 17 and 18: “Plasmalogens, a subclass of glycerophospholipids containing a vinyl-ether bond, are major components of biological membranes.”

Revised lines 17 and 18: “Plasmalogens, a subclass of glycerophospholipids containing a vinyl-ether bond, are one of the major components of biological membranes.”

Former lines 42 and 43: “Plasmalogens, a vinyl-ether subclass of glycerophospholipids, are major lipid components of biological membranes.”

Revised lines 42 and 43: “Plasmalogens, a vinyl-ether subclass of glycerophospholipids, are one of the major lipid components of biological membranes.”

Line 48: Please provide references.

Several references are provided in section III. Plasmalogen changes in pathophysiological conditions. We have reworded the text to indicate that (please see below).

Former lines 46-48: “Due to their high abundance, it is not unexpected that loss of plasmalogens has been associated with several pathologies ranging from inherited to metabolic and degenerative disorders.”

Revised lines 46-48: “Because of their high abundance, it is not unexpected that loss of plasmalogens has been associated with several pathologies ranging from inherited to metabolic and degenerative disorders (see section III below).”

Line 61: The authors emphasized the characteristics of "vinyl-ether" and showed the figure of PlsEtn, PakEtn, and PE. But as it is the first figure of the review, I recommend the authors consider all the types of plasmalogens, for example, PlsCho.

We have added the structure of plasmenylcholine (please see new figure).

Line 51~249: "A – Chemical structure", "B – Membrane physical properties", "C – Biological properties", "The metabolism of plasmalogens", "Plasmalogen changes in pathophysiological conditions". Frankly speaking, they have been summarized in previous reviews. So, it is not interesting to talk at length about them. Please compact the texts, and let your main point "PRT".

We made those sections more concise (please see below).

Former section I.A:

“A – Chemical structure

The chemical structure of plasmalogens is like their diacyl glycerophospholipids counterparts; that is, they have a

glycerol backbone, an acyl chain esterified to the position sn-2, and an alcohol linked to the position sn-3 of the glycerol group (the headgroup) (Figure 1). In mammals, the predominant acyl chains are the polyunsaturated fatty acids (PUFA) docosahexaenoic acid (DHA, 22:6, an w-3 fatty acid) and arachidonic acid (AA, 20:4, an w-6 fatty acid), while choline and ethanolamine are the two most common headgroups [5,6]. Plasmalogens differ from their diacyl counterparts by having an alkyl (instead of an acyl) chain attached via a vinyl-ether (instead of an ester) bond to the sn-1 position of the glycerol moiety (Figure 1). The alkyl chain in plasmalogens is, usually, saturated/monounsaturated and 16-18 carbon atoms long [5,7]. The presence of a vinyl-ether bond differs plasmalogen from other ether glycerophospholipids (Figure 1). In comparison to the ester bond, the vinyl-ether bond is more hydrophobic and acid/oxidation labile as well as less involved in hydrogen bonds [8]. These differences in the chemical properties lead to changes in the physical and biological properties of plasmalogens in comparison to their diacyl counterparts.”

Revised section I.A:

“A – Chemical structure

The chemical structure of plasmalogens is like their diacyl glycerophospholipids counterparts (Figure 1) [5,6,7]. Plasmalogens differ from their diacyl counterparts by having an alkyl (instead of an acyl) chain attached via a vinyl-ether (instead of an ester) bond to the sn-1 position of the glycerol moiety (Figure 1). The presence of a vinyl-ether bond makes plasmalogen different from other ether glycerophospholipids (i.e., plasmanyl phospholipids) (Figure 1). In comparison to the ester bond, the vinyl-ether bond is more hydrophobic and acid/oxidation labile as well as less involved in hydrogen bonds [8].”

Former section I.B:

“B – Membrane physical properties

The high abundance of plasmalogen in biological membranes together with differences in their chemical structures indicate that plasmalogens play a unique role in determining membrane physical properties. Indeed, it has been reported that plasmalogen imparts different physical properties to membranes in comparison to their diacyl counterparts. For instance, in comparison to their diacyl counterparts, plasmalogens tend to orient their headgroup more towards the water than the membrane-water interface as well as increasing the orientational ordering along the membrane normal of both hydrophobic chains [7,9–14]. This leads to an increased lipid packing and, consequently, increases thickness and decreases fluidity of the membrane. These properties tend to favor the stabilization of membrane domains, suggesting plasmalogens might be lipid components of these specialized membrane regions. As expected, plasmalogens have been reported to be enriched in membrane domains as well as stabilizing them [15–18]. Plasmalogens have also been suggested to contribute to the formation and stabilization of curved membrane surfaces. It has been shown that plasmalogens favor the formation of inverted lipid phases, supporting their propensity to form curved membrane surfaces [19–21]. Along these lines, plasmalogens have been shown to be enriched in curved membrane regions in cells (e.g., endoplasmic reticulum, ER, and Golgi cisterna) [22].”

Revised section I.B:

“B – Membrane physical properties

Plasmalogen due to their different chemical structure imparts different physical properties to membranes in comparison to their diacyl counterparts. For instance, plasmalogens tend to increase lipid packing and membrane thickness, decrease membrane fluidity, as well as contributing to the formation and stabilization membrane domains and curved membrane surfaces [7,9–22].”

Former section I.C:

“C – Biological properties

The fundamental relationship of biology states that the structure (and dynamics) of a molecule or molecular aggregate determines its function. Along this line, it is expected that plasmalogens, by having a different chemical structure, might have a different biological function compared to their diacyl counterparts. Indeed, this is what is found. For instance, one of the main biological functions ascribed to plasmalogens is their ability to function as scavengers of radical species such as reactive oxygen and nitrogen species (ROS/RNS) [23–28]. This leads to plasmalogens playing key

roles in the regulation of oxidative stress. Plasmalogen levels have been shown to inversely correlate with ROS levels in different model studies [28–30]. The protection against oxidative stress is ascribed to the presence of an oxidation labile vinyl-ether bond in plasmalogens. These lipids protect PUFA from ROS/RNS attack as well as terminate lipid peroxidation (the product of the oxidation of vinyl-ether bond is unable to propagate oxidation reactions) [29–32]. In addition, plasmalogens have been suggested to play key roles in signal transduction, including effects on the MAPK/ERK, PI3K/AKT, and PKCd pathways [33–36]. For instance, it has been shown that membrane plasmalogen play a role in the phagocytic ability of macrophages, a property that was related to the ability of plasmalogens to increase the number and size of lipid domains, which, in turn, improved the efficiency of phagocytic receptors-mediated intracellular signaling events [36,37]. Plasmalogens can also function by storing signaling molecules. For instance, oxidation of the vinyl-ether bond by hypochloric acid (derived from the reaction of the neutrophil myeloperoxidase) leads to the production of 2-chloro fatty aldehydes and fatty acids, which have been shown to modulate the inflammatory and immune responses to infection [38,39]. The hydrolysis of the acyl chain at the sn-2 position of the glycerol moiety in plasmalogens has also been suggested to modulate inflammatory processes [36]. This is due to the enrichment of PUFA at this position in plasmalogens and their role in mediating inflammation. More recently, plasmalogen gained increased interest in the study of treatment-resistant cancer due to their involvement in the regulation of lipid peroxidation and ferroptosis (a cell death process triggered by excessive lipid peroxidation) [40–42]. However, the molecular mechanisms underpinning the role of plasmalogens in lipid peroxidation and ferroptosis are not completely understood. Finally, plasmalogens have been suggested to play a role in membrane trafficking and viral infection [43–45]. In these processes the biological function of plasmalogens has been proposed to be a consequence of their ability to form and stabilize curved membrane regions, which, in turn, increase membrane remodeling needed during these biological phenomena.”

Revised section I.C:

“C – Biological properties

The fundamental relationship of biology states that the structure (and dynamics) of a molecule or molecular aggregate determines its function. Along this line, it is expected that plasmalogens, by having a different chemical structure, might have a different biological function compared to their diacyl counterparts. Indeed, this is what is found. For instance, one of the main biological functions ascribed to plasmalogens is their ability to function as scavengers of radical species such as reactive oxygen and nitrogen species (ROS/RNS) [23–32]. In addition, plasmalogens have been suggested to play key roles in signal transduction, including effects on the MAPK/ERK, PI3K/AKT, and PKCd pathways [33–37]. Plasmalogens can also function by storing signaling molecules as part of the structure of plasmalogens [36,38,39]. More recently, plasmalogen gained increased interest in the study of treatment-resistant cancer due to their involvement in the regulation of lipid peroxidation and ferroptosis (a cell death process triggered by excessive lipid peroxidation) [40–42]. However, the molecular mechanisms underpinning the role of plasmalogens in lipid peroxidation and ferroptosis are not completely understood. Finally, plasmalogens have been suggested to play a role in membrane trafficking and viral infection [43–45]. In these processes the biological function of plasmalogens has been proposed to be a consequence of their ability to form and stabilize curved membrane regions, which, in turn, increase membrane remodeling needed during these biological phenomena.”

Former section II:

“II. The metabolism of plasmalogens

Plasmalogens steady-state levels are a result of the difference between their rate of biosynthesis and of degradation. Plasmalogens biosynthesis starts in the peroxisomes and ends in the ER (Figure 2A). Contrary to the biosynthesis of diacyl glycerophospholipids, whose biosynthesis starts with glycerol-3-phosphate, plasmalogen biosynthesis starts with dihydroxyacetone phosphate (DHAP) [27,46,47]. In the lumen of the peroxisomes, DHAP is acylated by the activity of glyceronephosphate O-acyltransferase (GNPAT), which uses acyl-CoA as the acyl donor. GNPAT forms a heterotrimeric complex with alkyl-DHAP synthase (AGPS). The latter catalyzes the exchange of the acyl chain with a fatty alcohol leading to the production of 1-alkyl DHAP. The fatty alcohol comes either through the diet or by the activity of fatty acyl-CoA reductase, Far1. Far1 is an enzyme bound to the peroxisomal membrane external

(cytoplasmic) surface, which has been proposed to be rate-limiting reaction in plasmalogen biosynthesis [48–51]. Subsequently, the ketone in 1-alkyl DHAP is reduced by the activity of acyl/alkyl-DHAP reductase (AADHAP-R) leading to the production of 1-alkyl-2-lyso-sn-glycero-3-phosphate (AGP). AADHAP-R has been reported to be located at the external (cytoplasmic) surface of both peroxisomes and ER. In the ER, lysophosphatidate acyltransferase (AAG3P-AT) transfers an acyl chain (from acyl-CoA) to the position sn-2 of AGP yielding 1-alkyl-2-acyl-sn-glycerol-3-phosphate (plasmanyl-PA, an ether lipid without the vinyl-ether double bond) [27,46,47]. After the formation of plasmanyl-PA, its phosphate group is hydrolyzed by the activity of phosphatidate phosphohydrolase 1 (PAP-1) yielding 1-alkyl-2-acyl-sn-glycerol. Then, ethanolamine phosphotransferase (EPT) catalyzes the reaction to link phosphoethanolamine to the hydroxyl group at the sn-3 position of the glycerol moiety producing plasmanyl-PE. The final reaction is the production of the vinyl-ether bond by converting plasmanyl-PE to plasmenyl-PE (the PE form of plasmalogen, PE-Pls). In humans, this reaction is catalyzed by the plasmanylethanolamine desaturase 1 (PEDS1) [52]. The biosynthesis of the other major plasmalogen, plasmenyl-PC (the PC form of plasmalogen, PC-Pls), is less well-characterized, but it has been proposed to occur via headgroup transfer and/or remodeling of PE-Pls [27,47]. Therefore, most of the biochemical reactions are common to the biosynthesis of both these lipids.

Plasmalogen degradation could occur either by non-enzymatic or enzymatic biochemical reactions (Figure 2B). The non-enzymatic mechanisms of plasmalogen degradation are chemical in nature and depend on vinyl-ether bond oxidation or hydrolysis; that is, radical or acid attack removes the alkyl chain at the sn-1 position of the glycerol moiety [8,27,28]. The enzymatic mechanisms are dependent mainly on the action of phospholipases, each of which could present a different substrate specificity. For instance, phospholipase C and D catalysis both result in the hydrolysis of the headgroup, leading to the production of 1-alkenyl-2-acyl-sn-glycerol and 1-alkenyl-2-acyl-sn-phosphatidic acid, respectively [53,54]. On the other hand, phospholipase A2 catalysis results in the hydrolysis of the acyl chain at sn-2 position of the glycerol moiety [55–58]. In addition, it has been shown that cytochrome c upon oxidative stress can act as a plasmalogenase releasing the alkyl chain at the sn-1 position of the glycerol moiety [59].”

Revised section II:

“II. The metabolism of plasmalogens

Plasmalogens steady-state levels are a result of the difference between their rate of biosynthesis and of degradation. Plasmalogens biosynthesis starts in the peroxisomes and ends in the ER (Figure 2A) [27,46,47,52]. Contrary to the biosynthesis of diacyl glycerophospholipids, whose biosynthesis starts with glycerol-3-phosphate, plasmalogen biosynthesis starts with dihydroxyacetone phosphate (DHAP). In the peroxisomes, DHAP undergoes three sequential reactions to yield 1-alkyl-2-lyso-sn-glycero-3-phosphate (AGP), which is transported to the ER where the final biochemical reactions of plasmalogen biosynthesis take place. Fatty acyl-CoA reductase (Far1) is an enzyme bound to the peroxisomal membrane external (cytoplasmic) surface, which has been proposed to be rate-limiting reaction in plasmalogen biosynthesis [48–51].

Plasmalogen degradation could occur either by non-enzymatic or enzymatic biochemical reactions (Figure 2B). The non-enzymatic mechanisms of plasmalogen degradation are chemical in nature and depend on vinyl-ether bond oxidation or hydrolysis; that is, radical or acid attack removes the alkyl chain at the sn-1 position of the glycerol moiety [8,27,28]. The enzymatic mechanisms are dependent mainly on the action of phospholipases, each of which could present a different substrate specificity [53-58]. In addition, it has been shown that cytochrome c upon oxidative stress can act as a plasmalogenase releasing the alkyl chain at the sn-1 position of the glycerol moiety [59].”

Former section III:

“III. Plasmalogen changes in pathophysiological conditions

Historically, plasmalogens had received little attention compared to various other lipid classes despite their abundance. However, recently this has changed and plasmalogens have started to receive increased attention. This is because of the association between plasmalogens and several pathophysiological conditions. It has been reported that

plasmalogen levels are altered in several degenerative and metabolic disorders as well as upon aging. In all these conditions, a common observation is the decrease in the levels of plasmalogens. For instance, humans early in life have a small content of plasmalogens in the brain, but by the first year of age the content of plasmalogen increases markedly (8-fold) [60]. Plasmalogens content keep increasing gradually up to 40 years of age, after which it tends to level off and, by the age of 70 plasmalogen content starts to decrease significantly (e.g., there is a 40% decrease of plasmalogen in the serum of individuals more than 70 years old in comparison to younger individuals) [61,62].

One of the first associations between plasmalogens and diseases came from studies of peroxisomal deficiency diseases. These are a collection of rare inherited human diseases caused by mutations in genes responsible for peroxisomes biogenesis or function. These diseases include Zellweger syndrome (ZS, an inherited disorder caused by a reduction or absence of functional peroxisomes and characterized by degeneration of brain white matter) and Rhizomelic chondrodysplasia punctata (RCDP, an inherited disorder caused by mutations in peroxisomal enzymes involved in plasmalogen biosynthesis and characterized by shortening of proximal bones) [63,64]. In these diseases plasmalogen levels are decreased. For instance, in ZS plasmalogens content has been found to be largely decreased, the extent of which is tissue-specific and can reach up to 90% decrease in comparison to controls [65]. In RCDP it has been reported that the decrease in plasmalogen varies with the severity of the phenotype, reaching up to more than 70% reduction in plasmalogen in the most severe phenotype [66].

Decreases in plasmalogen content have also been reported in degenerative and metabolic disorders. In the brain, where the plasmalogen content is the highest, plasmalogen loss has been reported in samples from individuals with different neurodegenerative disorders including Alzheimer’s disease (AD, a neurodegenerative disorder characterized by the presence amyloid-b plaques), Parkinson’s disease (PD, a neurodegenerative disorder characterized by the presence of fibrillar a-synuclein aggregates), and Multiple Sclerosis (a chronic auto-immune neurodegenerative disorder characterized by neuronal demyelination) [67–69]. For instance, in brain samples (post-mortem) of individuals with AD plasmalogen levels have been reported to be decreased 30-40%, while in blood samples from PD patients PE-ether lipids (including plasmalogens) showed a decrease of ca. 30% in comparison to healthy individuals [70,71]. Plasmalogen loss has also been reported in cardiometabolic diseases such as Barth Syndrome (BTHS, an inherited genetic disorder caused by mutations in the mitochondrial transacylase tafazzin and characterized by cardiomyopathy and skeletal myopathy as well as growth delay and neutropenia) and coronary artery diseases (CAD, the major form of heart diseases, which is characterized by the presence of lipid deposits namely atherosclerotic plaques inside arterial walls) [72,73]. While BTHS is a disease where the metabolism of cardiolipin is directly affected, new evidence points to an interrelationship between the metabolisms of cardiolipin and plasmalogens in BTHS [74]. It has been shown that plasmalogen levels are decreased significantly in BTHS. This was shown in different organs (brain, heart, and liver) of a tafazzin knockdown mouse as well in lymphoblast cells derived from BTHS patients [75,76]. In the case of CAD, plasmalogen levels were reported to be decreased in the plasma of patients with CAD in comparison to healthy individual controls [73,77].

In the blood, plasmalogens are found within erythrocyte membranes and lipoproteins [78]. Changes in blood plasmalogens have attracted some interest as a potential biomarker for the diagnosis and prognosis of some pathological diseases [79–85]. However, in all these cases comparisons were done with healthy individual controls. In addition, plasmalogen loss has been reported in various diseases (see above) and there is no disease-specific marker reported yet. Hence, changes in blood plasmalogens by itself should be viewed with caution. A better potential is to use blood plasmalogen changes with other biomarkers to enhance diagnosis/prognosis accuracy. For instance, it has been shown that changes in PC-Pls containing oleic acid at the sn-2 position of the glycerol moiety together with adiponectin and HDL-cholesterol (risk factors for atherosclerosis) in the serum might increase the identification of a proatherogenic state [86].”

Revised section III:

“III. Plasmalogen changes in pathophysiological conditions

Historically, plasmalogens had received little attention compared to various other lipid classes despite their abundance. However, recently this has changed and plasmalogens have started to receive increased attention. This is because of the association between plasmalogens and several pathophysiological conditions. It has been reported that

plasmalogen levels are altered in several degenerative and metabolic disorders as well as upon aging. In all these conditions, a common observation is the decrease in the levels of plasmalogens.

In humans plasmalogens content increases gradually up to 40 years of age, after which it tends to level off and, by the age of 70 plasmalogen content starts to decrease significantly (e.g., there is a 40% decrease of plasmalogen in the serum of individuals more than 70 years old in comparison to younger individuals) [60-62]. One of the first associations between plasmalogens and diseases came from studies of peroxisomal deficiency diseases. These are a collection of rare inherited human diseases caused by mutations in genes responsible for peroxisomes biogenesis or function, which include Zellweger syndrome (ZS) and Rhizomelic chondrodysplasia punctata (RCDP) [63,64]. In these diseases plasmalogen levels are decreased. For instance, in ZS plasmalogens content has been found to be largely decreased, the extent of which is tissue-specific and can reach up to 90% decrease in comparison to controls [65]. In RCDP it has been reported that the decrease in plasmalogen varies with the severity of the phenotype, reaching up to more than 70% reduction in plasmalogen in the most severe phenotype [66]. Decreases in plasmalogen content have also been reported in degenerative and metabolic disorders. In the brain, where the plasmalogen content is the highest, plasmalogen loss has been reported in samples from individuals with different neurodegenerative disorders including Alzheimer’s disease (AD), Parkinson’s disease (PD), and Multiple Sclerosis (MS) [67–71]. Plasmalogen loss has also been reported in cardiometabolic diseases such as Barth Syndrome (BTHS) and coronary artery diseases (CAD) [72-77].

In the blood, plasmalogens are found within erythrocyte membranes and lipoproteins [78]. Changes in blood plasmalogens have attracted some interest as a potential biomarker for the diagnosis and prognosis of some pathological diseases [79–85]. However, in all these cases comparisons were done with healthy individual controls. In addition, plasmalogen loss has been reported in various diseases (see above) and there is no disease-specific marker reported yet. Hence, changes in blood plasmalogens by itself should be viewed with caution. A better criterion is to use blood plasmalogen changes with other biomarkers to enhance diagnosis/prognosis accuracy. For instance, it has been shown that changes in PC-Pls containing oleic acid at the sn-2 position of the glycerol moiety together with adiponectin and HDL-cholesterol (risk factors for atherosclerosis) in the serum might increase the identification of a proatherogenic state [86].”

Line 220: Please make the "background" concise. The authors do not need to explain "what is AD/PD", especially, there is an unknown vortex symbol after the word "amyloid" and before the word "synuclein" (and otherwhere in the whole manuscript).

We have addressed this point together with previous one (please see new text above).

Line 252: What is the definition of "replacement", and what are the differences between "replacement therapy " and simply "therapy/treatment"? Oral administration could not prove it connected with "replacement". I strongly suggest the authors be very careful about using this word unless there is strong evidence.

Replacement therapy is a treatment (medical care given to an individual with a disease/injury) aimed at making up a deficit of a biological molecule normally present in the body. One example of replacement therapy is hormone replacement therapy, where women in menopause are given hormones to replace the estrogen that is no longer produced in the body, that is, to increase its levels.

We provide several references showing different strategies to replace the deficit in plasmalogens. While we agree with the hypothesis provided by the reviewer in the last comment (see below), there is strong evidence indicating that the strategies used lead to the compounds being metabolized and incorporated into plasmalogens and, consequently, increasing plasmalogen levels. Hence, we believe that is appropriated to use the term plasmalogen replacement therapy.

We acknowledge that without a formal definition this could be misunderstood. Therefore, we have changed the text to clarify its definition and avoid misunderstandings (please see new text of section IV. Plasmalogen Replacement Therapy Section (PRT) below. This part is highlighted in green to facilitate the identification).

Line 265: What are "plasmalogen precursors"? I guess the authors did not explain it until Line 288.

We have changed the text to clarify the meaning of plasmalogen precursors (please see new text below).

Former lines 264-267: “PRT could be implemented by dietary intervention. For instance, plasmalogens and plasmalogen precursors have been found to be enriched in marine animals (e.g., shark liver, krill, mussels, sea squirt/urchin/cucumber, scallops) as well as in land animals’ meat (e.g., pork, beef, and chicken) (Figure 3) [91].”

Revised lines 264-267: “PRT could be implemented by dietary intervention. For instance, plasmalogens and plasmalogen precursors (intermediates of the plasmalogen biosynthesis pathway) have been found to be enriched in marine animals (e.g., shark liver, krill, mussels, sea squirt/urchin/cucumber, scallops) as well as in land animals’ meat (e.g., pork, beef, and chicken) (Figure 3) [91].”

Line 252: If possible, please separate this part into several sub-sections, making the main body of this review a "stronger sense of layers".

We have changed this section into several sub-sections (please see below).

Former section IV “Plasmalogen replacement therapy (PRT)

A modern and innovative pharmacological approach that started to emerge is membrane lipid replacement [87,88]. Lately this strategy has attracted increased interest as potentially useful in a variety of pathological conditions including cancer, neurological, and metabolic disorders [89]. The idea behind membrane-lipid therapy is to develop drugs that target membrane lipids. Plasmalogen replacement therapy (PRT) is a type of membrane-lipid therapy where the strategy relies on the use of small molecules to increase plasmalogen levels with the final goal of improving health outcomes. One of the main advantages of PRT is the possibility to use oral administration. Another one is that the compounds used in PRT, usually, exhibit no toxicity even at high doses and have been reported to be safe for use in humans [90].

PRT could be implemented by dietary intervention. For instance, plasmalogens and plasmalogen precursors (intermediates of plasmalogen biosynthesis pathway) have been found to be enriched in marine animals (e.g., shark liver, krill, mussels, sea squirt/urchin/cucumber, scallops) as well as in land animals’ meat (e.g., pork, beef, and chicken) (Figure 3) [91]. It has been shown that plasmalogens levels are higher (ranging from 2-50-fold, depending on the exact comparison) in livestock and poultry than in fish and mollusk [91]. However, an interesting finding is that the

plasmalogens from fish and mollusk have a lower w-6/w-3 fatty acid ratio than livestock ones, suggesting the former provide an advantage because of the proposed health benefits of w-3 fatty acids. While natural sources of plasmalogens bear the potential of providing a dietary plasmalogen supplementation, the decreased bioavailability and the enormous amount of raw material required, makes it impractical. For instance, scallops have ca. 7.5 mg of plasmalogen/g of muscle. To achieve a common dose of 50 mg/kg, it means that a human with an average weight of 70 kg would need to eat ca. 460 kg of scallops.

To circumvent these problems purified or synthetic compounds are an attractive alternative to implement PRT as they can be administered at a high dosage. One possibility is to use plasmalogen extracts from natural sources. These plasmalogen extracts tend to be enriched in PE-Pls and they are often prepared from marine animals (e.g., scallop and sea squirt) or from chicken (Figure 3) [91]. Another possibility is the use of plasmalogen precursors, such as alkylglycerols (AG). AG are lipid intermediates of the plasmalogen biosynthesis pathway (see above) that readily cross the cellular plasma membrane and can enter as a component of the plasmalogen biosynthesis pathway in the ER, after being phosphorylated in the cytosol (Figure 2) [2,92]. AG of different chain lengths have been used, the most common ones being 1-O-hexadecyl-sn-glycerol (HG, 16:0-AG), 1-O-octadecyl-sn-glycerol (OG, 18:0-AG), and 1-O-octadecenyl-sn-glycerol (OeG, 18:1-AG) (Figure 4) [3]. In mammals, oral administration of purified plasmalogens shows extensive breakdown in the intestinal mucosal cells, while administration of AG leads to complete absorption by the intestine without cleavage of the ether bond; likely because the vinyl-ether bond in plasmalogens is more acid/oxidation labile than the ether bond in AG [93]. In the intestinal mucosal cells, the majority of AG is metabolized into plasmalogens (specifically PE-Pls), while a small fraction is transported to the liver where it is metabolized [90]. Plasmalogens are transported from the intestine and liver to other organs, but it seems they do not cross the blood-brain barrier and are not transported across the placenta (from mother to fetus) [90,94]. The use of synthetic analogs of plasmalogens in PRT has also been described, e.g., PPI-1011 (an alkyl-diacyl plasmalogen precursor with DHA at the sn-2 position), PPI-1025 (an alkyl-diacyl plasmalogen precursor with oleoyl at the sn-2 position), and PPI-1040 (a PE-Pls analogue with a proprietary cyclic PE headgroup) (Figure 4) [95–97]. PPI-1011 is bioavailable in rabbits and leads to an increase in PE-Pls levels in circulation. In addition, its metabolites can cross both the blood-retina and the blood-brain barriers [95]. PPI-1040 leads to a greater increase in plasmalogen level and bioavailability than PPI-1011 [97]. Indeed, contrary to PPI-1011, PPI-1040 has been reported to be intact during digestion, absorption, and circulation, possibly explaining its greater efficacy [95,97].

PRT has been studied in different clinical settings from cells to animals and humans (Table 1). In cells, PRT has been shown to be successful in increasing plasmalogen levels and alleviating some disease-related phenotypes (Table 1). For instance, in lymphoblasts-derived from BTHS patients, adding HG to the media 20 h before collecting the cells led to a significant increase in PE-Pls levels [98]. In this study an increase in cardiolipin levels as well as an improvement in mitochondrial fitness were also found, indicating the potential benefit of PRT to improve health outcomes of BTHS patients. For ZS, it was shown that administration of HG to fibroblasts-derived from ZS subjects, results in increased PE-Pls levels and decreased b-adrenergic receptor stimulation by isoproterenol (an agonist), a result of a reduced number of receptors induced by HG treatment [99]. In lymphocytes-derived from RCDP patients, administration of PPI-1011 increased PE-Pls levels [95]. In addition, administration of purified PE-Pls to neurons promoted differentiation with the greatest effect coming from PE-Pls purified from a marine mollusk (M. edulis) rather than bovine brain, possibly due to differences in lipid molecular species [100]. Scallop-purified PE-Pls was shown to have anti-inflammatory properties as indicated by reduction of microglia activation, toll-like receptor 4 (TLR4) endocytosis, and caspase activation [101]. In neurons, chicken purified PE-Pls has anti-apoptotic properties as indicated by a decrease in caspase activation and activation of PI3K/AKT and MAPK/ERK signaling pathways [102]. Likewise, eicosapentaenoic acid (EPA)-enriched PE-Pls also showed anti-apoptotic properties in primary cultured hippocampal neurons by upregulating anti-apoptotic proteins and downregulating apoptotic proteins [103]. In addition, in an isolated rat heart, administration of HG decreased myocardial ischemia/reperfusion injury [30].

In animals, PRT was also successful in increasing plasmalogen levels and ameliorating some disease-related phenotypes (Table 1). In a mouse model of RCDP (GNPAT knockout), OG administration for 2 months replenishes cardiac levels of PE-Pls and normalized cardiac impulse [104]. In another mouse model of RCDP (Pex7 knockout), OG increased PE-Pls levels in peripheral tissues and nervous tissue as well as improving nerve conduction [105]. In this

study, PRT stopped the progression of pathology in testis, adipose tissue, and eyes. In the same Pex7 knockout mice, administration of PI-1040 increased levels of PE-Pls in the plasma, erythrocytes, liver, small intestine, skeletal muscle, and heart, but not in brain, lung, and kidney [97]. In this study, PRT reduced the hyperactive behavior of Pex7 knockout mice. Interestingly, PPI-1011 did not show the same effects as PI-1040 treatment, suggesting that the latter is a better candidate for use in future investigations of RCDP. Indeed, in 2019 the FDA (Food and Drug Administration) granted PPI-1040 orphan drug designation for treatment of RCDP.

There has also been some interest in the use of PRT in animal models of neurological and metabolic disorders (Table 1). For instance, in mice models of AD, oral administration or intraperitoneal injection of purified PE-Pls led to neuroprotection by attenuating neurotoxicity and neuroinflammation in the brain as indicated by a reduced microglia activation, reduced accumulation of Ab peptides, decreased neuronal apoptosis and activation of brain-derived neurotrophic factor/tropomyosin receptor B/cAMP response element-binding protein (BDNF/TrkB/CREB) signaling pathway, and inhibition of oxidative stress [101,103,106,107]. In addition, in cognitive-deficient rats (a model for AD), oral administration of purified PE-Pls led to increase in PE-Pls levels in the plasma, erythrocyte, and liver as well as improved reference/working memory-related learning abilities [108]. While an increase in total levels was not observed in the brain, the cerebral cortex became enriched with the major molecular species of the purified PE-Pls, that is, 18:0/22:6-PE-Pls. In rats the co-administration of myo-inositol and ethanolamine has been shown to increase cerebellum PE-Pls levels as well as decrease cortex ATP and cerebellar oxidative stress [120,121]. In animal (mouse and monkey) models of PD (treated with 1-methyl-4-phenyl-1,2,3,6-tetrahydropyridine, MPTP), oral administration of PPI-1011 increased PE-Pls levels in serum, displayed neuroprotective and anti-inflammatory properties, reversed dopamine and serotonin loss as well as showed antidyskinetic activity [71,96,109–111,122]. A similar result was obtained with PPI-1025 treatment, suggesting that the effect is independent of the acyl chain at the sn-2 position of the glycerol moiety and likely dependent on the vinyl-ether bond or on the entire plasmalogen molecule [96]. In hamster and mouse models of atherosclerosis, oral administration of PE-Pls decreased atherosclerosis lesion and total cholesterol and LDL cholesterol in serum [112,123]. In transgenic mice with small (due to depressed PI3K signaling) or failing hearts (due to dilated cardiomyopathy), OG treatment increased both PC-Pls and PE-Pls levels but did not impact on heart size or function [113]. PRT had also shown antitumor properties. In mice, oral administration of shark liver oil (enriched in AG) or shark-liver oil-purified AG decreased tumor growth, vascularization, and dissemination [114].

In humans, PRT has been investigated in subjects with peroxisomal, neurodegenerative, and metabolic disorders (Table 1). Two independent early studies of PRT with humans, showed that oral administration of OG to individuals with peroxisomal disorders (characterized by decreased GNPAT activity and decreased levels of PE-Pls in erythrocyte membranes) led to restoration of erythrocyte PE-Pls content and clinical (growth, muscle/motor tone, general state of awareness, liver function, retinal pigmentation) improvement [90,115]. In an intention-to-treat analysis, oral administration of scallop-purified PE-Pls to individuals with mild AD and mild cognitive impairment did not show a significant difference from the control (placebo) group in primary (Mini mental state examination-Japanese, MMSE-J) or secondary (Wechsler Memory Scale-revised, WMS-R, Geriatric depression scale-short version-Japanese, GDSSV-J, and plasma erythrocyte PE-Pls levels) outcomes [116]. However, in another study, oral administration of scallop-purified PE-Pls to individuals with mild cognitive impairment, mild-to-severe AD, and PD led to significant increase in plasma and erythrocyte PE-Pls levels as well as cognitive function [124]. Likewise, in subjects with mild forgetfulness, sea squirt-purified PE-Pls showed improved cognitive function as evaluated by visual and verbal memory scores [117]. Individuals with PD treated with scallop-purified PE-Pls showed increased PE-Pls content in the plasma and erythrocytes membranes [71]. Improvement was also noted in PD clinical symptoms as evaluated by Parkinson’s disease questionnaire-39 (PDQ-39, a self-report questionnaire used to monitor health status of physical, mental, and social domains). In individuals with features of metabolic disease, oral administration of shark liver oil-purified AG led to increased levels of PE-Pls in plasma and circulatory white blood cells as well as decreased total cholesterol, triglycerides, and C-reactive protein in the plasma [118]. In subjects with hyperlipidemia, myo-inositol treatment led to increase in PC-Pls and decrease in atherogenic cholesterol in the serum, suggesting the potential benefit for individuals with metabolic syndrome [119].”

Revised section IV “Plasmalogen replacement therapy (PRT)

A modern and innovative pharmacological approach that started to emerge is membrane lipid replacement [87,88]. A replacement therapy is a pharmacological intervention aimed at restoring the levels of a biological molecule that is deficient in some pathophysiological condition. Lately this strategy has attracted increased interest as potentially useful in a variety of pathological conditions including cancer, neurological, and metabolic disorders [89]. Plasmalogen replacement therapy (PRT) is a type of membrane lipid replacement where the strategy relies on the use of small molecules to increase plasmalogen levels with the final goal of improving health outcomes. One of the main advantages of PRT is the possibility to use oral administration. Another one is that the compounds used in PRT, usually, exhibit no toxicity even at high doses and have been reported to be safe for use in humans [90].

A. Small molecules used in PRT

PRT could be implemented by dietary intervention. For instance, plasmalogens and plasmalogen precursors (intermediates of plasmalogen biosynthesis pathway) have been found to be enriched in marine animals (e.g., shark liver, krill, mussels, sea squirt/urchin/cucumber, scallops) as well as in land animals’ meat (e.g., pork, beef, and chicken) (Figure 3) [91]. It has been shown that plasmalogens levels are higher (ranging from 2-50-fold, depending on the exact comparison) in livestock and poultry than in fish and mollusk [91]. However, an interesting finding is that the plasmalogens from fish and mollusk have a lower w-6/w-3 fatty acid ratio than livestock ones, suggesting the former provide an advantage because of the proposed health benefits of w-3 fatty acids. While natural sources of plasmalogens bear the potential of providing a dietary plasmalogen supplementation, the decreased bioavailability and the enormous amount of raw material required, makes it impractical. For instance, scallops have ca. 7.5 mg of plasmalogen/g of muscle. To achieve a common dose of 50 mg/kg, it means that a human with an average weight of 70 kg would need to eat ca. 460 kg of scallops.

To circumvent these problems purified or synthetic compounds are an attractive alternative to implement PRT as they can be administered at a high dosage. One possibility is to use plasmalogen extracts from natural sources. These plasmalogen extracts tend to be enriched in PE-Pls and they are often prepared from marine animals (e.g., scallop and sea squirt) or from chicken (Figure 3) [91]. Another possibility is the use of plasmalogen precursors, such as alkylglycerols (AG). AG are lipid intermediates of the plasmalogen biosynthesis pathway (see above) that readily cross the cellular plasma membrane and can enter as a component of the plasmalogen biosynthesis pathway in the ER, after being phosphorylated in the cytosol (Figure 2) [2,92]. AG of different chain lengths have been used, the most common ones being 1-O-hexadecyl-sn-glycerol (HG, 16:0-AG), 1-O-octadecyl-sn-glycerol (OG, 18:0-AG), and 1-O-octadecenyl-sn-glycerol (OeG, 18:1-AG) (Figure 4) [3]. In mammals, oral administration of purified plasmalogens shows extensive breakdown in the intestinal mucosal cells, while administration of AG leads to complete absorption by the intestine without cleavage of the ether bond; likely because the vinyl-ether bond in plasmalogens is more acid/oxidation labile than the ether bond in AG [93]. In the intestinal mucosal cells, the majority of AG is metabolized into plasmalogens (specifically PE-Pls), while a small fraction is transported to the liver where it is metabolized [90]. Plasmalogens are transported from the intestine and liver to other organs, but it seems they do not cross the blood-brain barrier and are not transported across the placenta (from mother to fetus) [90,94]. The use of synthetic analogs of plasmalogens in PRT has also been described, e.g., PPI-1011 (an alkyl-diacyl plasmalogen precursor with DHA at the sn-2 position), PPI-1025 (an alkyl-diacyl plasmalogen precursor with oleoyl at the sn-2 position), and PPI-1040 (a PE-Pls analogue with a proprietary cyclic PE headgroup) (Figure 4) [95–97]. PPI-1011 is bioavailable in rabbits and leads to an increase in PE-Pls levels in circulation. In addition, its metabolites can cross both the blood-retina and the blood-brain barriers [95]. PPI-1040 leads to a greater increase in plasmalogen level and bioavailability than PPI-1011 [97]. Indeed, contrary to PPI-1011, PPI-1040 has been reported to be intact during digestion, absorption, and circulation, possibly explaining its greater efficacy [95,97].

B. In vitro PRT studies

PRT has been studied in different clinical settings from cells to animals and humans (Table 1). In cells, PRT has been shown to be successful in increasing plasmalogen levels an

For instance, in lymphoblasts-derived from BTHS patients, adding HG to the media 20 h before collecting the cells led to a significant increase in PE-Pls levels [98]. In this study an increase in cardiolipin levels as well as an improvement in mitochondrial fitness were also found, indicating the potential benefit of PRT to improve health outcomes of BTHS patients. For ZS, it was shown that administration of HG to fibroblasts-derived from ZS subjects, results in increased PE-Pls levels and decreased b-adrenergic receptor stimulation by isoproterenol (an agonist), a result of a reduced number of receptors induced by HG treatment [99]. In lymphocytes-derived from RCDP patients, administration of PPI-1011 increased PE-Pls levels [95]. In addition, administration of purified PE-Pls to neurons promoted differentiation with the greatest effect coming from PE-Pls purified from a marine mollusk (M. edulis) rather than bovine brain, possibly due to differences in lipid molecular species [100]. Scallop-purified PE-Pls was shown to have anti-inflammatory properties as indicated by reduction of microglia activation, toll-like receptor 4 (TLR4) endocytosis, and caspase activation [101]. In neurons, chicken purified PE-Pls has anti-apoptotic properties as indicated by a decrease in caspase activation and activation of PI3K/AKT and MAPK/ERK signaling pathways [102]. Likewise, eicosapentaenoic acid (EPA)-enriched PE-Pls also showed anti-apoptotic properties in primary cultured hippocampal neurons by upregulating anti-apoptotic proteins and downregulating apoptotic proteins [103]. In addition, in an isolated rat heart, administration of HG decreased myocardial ischemia/reperfusion injury [30].

C. In vivo PRT studies

In animals, PRT was also successful in increasing plasmalogen levels and ameliorating some disease-related phenotypes (Table 1). In a mouse model of RCDP (GNPAT knockout), OG administration for 2 months replenishes cardiac levels of PE-Pls and normalized cardiac impulse [104]. In another mouse model of RCDP (Pex7 knockout), OG increased PE-Pls levels in peripheral tissues and nervous tissue as well as improving nerve conduction [105]. In this study, PRT stopped the progression of pathology in testis, adipose tissue, and eyes. In the same Pex7 knockout mice, administration of PI-1040 increased levels of PE-Pls in the plasma, erythrocytes, liver, small intestine, skeletal muscle, and heart, but not in brain, lung, and kidney [97]. In this study, PRT reduced the hyperactive behavior of Pex7 knockout mice. Interestingly, PPI-1011 did not show the same effects as PI-1040 treatment, suggesting that the latter is a better candidate for use in future investigations of RCDP. Indeed, in 2019 the FDA (Food and Drug Administration) granted PPI-1040 orphan drug designation for treatment of RCDP.

There has also been some interest in the use of PRT in animal models of neurological and metabolic disorders (Table 1). For instance, in mice models of AD, oral administration or intraperitoneal injection of purified PE-Pls led to neuroprotection by attenuating neurotoxicity and neuroinflammation in the brain as indicated by a reduced microglia activation, reduced accumulation of Ab peptides, decreased neuronal apoptosis and activation of brain-derived neurotrophic factor/tropomyosin receptor B/cAMP response element-binding protein (BDNF/TrkB/CREB) signaling pathway, and inhibition of oxidative stress [101,103,106,107]. In addition, in cognitive-deficient rats (a model for AD), oral administration of purified PE-Pls led to increase in PE-Pls levels in the plasma, erythrocyte, and liver as well as improved reference/working memory-related learning abilities [108]. While an increase in total levels was not observed in the brain, the cerebral cortex became enriched with the major molecular species of the purified PE-Pls, that is, 18:0/22:6-PE-Pls. In rats the co-administration of myo-inositol and ethanolamine has been shown to increase cerebellum PE-Pls levels as well as decrease cortex ATP and cerebellar oxidative stress [120,121]. In animal (mouse and monkey) models of PD (treated with 1-methyl-4-phenyl-1,2,3,6-tetrahydropyridine, MPTP), oral administration of PPI-1011 increased PE-Pls levels in serum, displayed neuroprotective and anti-inflammatory properties, reversed dopamine and serotonin loss as well as showed antidyskinetic activity [71,96,109–111,122]. A similar result was obtained with PPI-1025 treatment, suggesting that the effect is independent of the acyl chain at the sn-2 position of the glycerol moiety and likely dependent on the vinyl-ether bond or on the entire plasmalogen molecule [96]. In hamster and mouse models of atherosclerosis, oral administration of PE-Pls decreased atherosclerosis lesion and total cholesterol and LDL cholesterol in serum [112,123]. In transgenic mice with small (due to depressed PI3K signaling) or failing hearts (due to dilated cardiomyopathy), OG treatment increased both PC-Pls and PE-Pls levels but did not impact on heart size or function [113]. PRT had also shown antitumor properties. In mice, oral administration of shark liver oil (enriched in AG) or shark-liver oil-purified AG decreased tumor growth, vascularization, and dissemination [114].

D. Clinical trials

In humans, PRT has been investigated in subjects with peroxisomal, neurodegenerative, and metabolic disorders (Table 1). Two independent early studies of PRT with humans, showed that oral administration of OG to individuals with peroxisomal disorders (characterized by decreased GNPAT activity and decreased levels of PE-Pls in erythrocyte membranes) led to restoration of erythrocyte PE-Pls content and clinical (growth, muscle/motor tone, general state of awareness, liver function, retinal pigmentation) improvement [90,115]. In an intention-to-treat analysis, oral administration of scallop-purified PE-Pls to individuals with mild AD and mild cognitive impairment did not show a significant difference from the control (placebo) group in primary (Mini mental state examination-Japanese, MMSE-J) or secondary (Wechsler Memory Scale-revised, WMS-R, Geriatric depression scale-short version-Japanese, GDSSV-J, and plasma erythrocyte PE-Pls levels) outcomes [116]. However, in another study, oral administration of scallop-purified PE-Pls to individuals with mild cognitive impairment, mild-to-severe AD, and PD led to significant increase in plasma and erythrocyte PE-Pls levels as well as cognitive function [124]. Likewise, in subjects with mild forgetfulness, sea squirt-purified PE-Pls showed improved cognitive function as evaluated by visual and verbal memory scores [117]. Individuals with PD treated with scallop-purified PE-Pls showed increased PE-Pls content in the plasma and erythrocytes membranes [71]. Improvement was also noted in PD clinical symptoms as evaluated by Parkinson’s disease questionnaire-39 (PDQ-39, a self-report questionnaire used to monitor health status of physical, mental, and social domains). In individuals with features of metabolic disease, oral administration of shark liver oil-purified AG led to increased levels of PE-Pls in plasma and circulatory white blood cells as well as decreased total cholesterol, triglycerides, and C-reactive protein in the plasma [118]. In subjects with hyperlipidemia, myo-inositol treatment led to increase in PC-Pls and decrease in atherogenic cholesterol in the serum, suggesting the potential benefit for individuals with metabolic syndrome [119].”

Page 17 (by the way, there is no line number), "V. Future perspective for PRT": I still feel it is not enough to claim "plasmalogen replacement".

Suppose, "plasmalogens are oxidants and easy to be oxidized, the anti-oxidative effect of exogenous plasmalogens improves the body conditions, and the endogenous plasmalogens are protected against oxidative stress" -> this is a possible theory, but cannot be described as "replacement"!

We have addressed this at comments 6 and 8 (please see above).

Round 2

Reviewer 2 Report

The authors have responded the questions and comments, making appropriate modifications and revisions. The manuscript quality is improved, which is suitable for acceptance.